# EgoDistill: Egocentric Head Motion Distillation for Efficient Video Understanding

**Shuhan Tan**[1], **Tushar Nagarajan**[2], **Kristen Grauman**[1,2]
[1]University of Texas at Austin, [2]FAIR, Meta

## Abstract

Recent advances in egocentric video understanding models are promising, but their heavy computational expense is a barrier for many real-world applications. To address this challenge, we propose EgoDistill, a distillation-based approach that learns to reconstruct heavy egocentric video clip features by combining the semantics from a sparse set of video frames with the head motion from lightweight IMU readings. We further devise a novel self-supervised training strategy for IMU feature learning. Our method leads to significant improvements in efficiency, requiring $200\times$ fewer GFLOPs than equivalent video models. We demonstrate its effectiveness on the Ego4D and EPIC-Kitchens datasets, where our method outperforms state-of-the-art efficient video understanding methods. Project page: https://vision.cs.utexas.edu/projects/egodistill/

## 1   Introduction

Recent advances in augmented and virtual reality (AR/VR) technology have the potential to change the way people interact with the digital world, much like the smartphone did in the previous decade. A fundamental requirement for AR/VR systems is the ability to recognize user behavior from egocentric video captured from a head-mounted camera. Towards this goal, several egocentric video datasets have been proposed in recent years, spurring increasing attention of the research community [26, 56, 11].

Current egocentric video understanding models use powerful *clip-based* video backbones that operate on video clips of a few seconds at a time [18, 54, 25, 16, 12, 55, 43, 44]. Despite encouraging performance, these models typically process densely sampled frames with temporally-aware operations, making them computationally heavy. This makes them impractical for AR/VR devices with constrained resources, or for real-time video applications that require low latency. How to efficiently perform egocentric video understanding is therefore an important, yet unsolved problem.

To address this issue, we take inspiration from how animals perceive the world with ego-motion. Neuroscience research has found that during active movement, the animal visual cortex receives and encodes head motion signals from the motor cortex for visual processing [27, 52, 53], highlighting its role in the efficient understanding of the animal's visual stream. Inspired by this phenomenon, we explore the relationship between human head motion and ego-video for efficient video understanding.

In practice, we consider head motion signals captured by the inertial measurement unit (IMU) of a head-mounted camera. IMU measures motion from an accelerometer and gyroscope and is widely available on popular wearable devices. Prior work leverages IMU as an extra modality for human action recognition [68, 69, 13] (e.g., jumping, walking), or as geometric cues for visual-inertial odometry [7, 20, 71]. In contrast, we propose to drive efficient video understanding by drawing on IMU as a *substitute* for dense video frame observations. The intuition is as follows. A video clip contains two things: semantic content (appearance of objects, places, people) and dynamics (how the scene and the camera move). While densely sampled frames are sure to capture both of the above—as done by current clip models [54, 16, 17]—we hypothesize they are sometimes overkill. For a short

37th Conference on Neural Information Processing Systems (NeurIPS 2023).

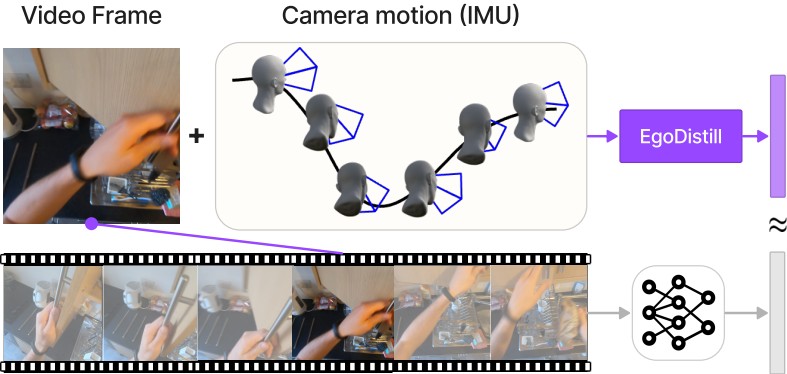

Figure 1: **Illustration of EgoDistill.** Given a single video frame and camera motion from IMU, EgoDistill learns to reconstruct the more expensive dense video clip feature. With its lightweight input, EgoDistill significantly improves efficiency.

video clip, much of the semantic content is intelligible from even a single frame; meanwhile, the head motion provides a good portion of the dynamics, implicitly revealing how the visual appearance changes across neighboring frames.

Building on this insight, we introduce EgoDistill, an approach that learns to reconstruct dense egocentric video clip features using temporally sparse visual observations (as few as one RGB frame) together with the head motion from IMU. Specifically, EgoDistill employs a new form of knowledge distillation from video models. During training, we train a lightweight model that takes sparsely sampled image(s) and IMU to approximate video features extracted by a powerful but expensive video model. We further improve the model with a novel IMU-guided self-supervised training stage. During inference, we directly utilize the lightweight model for egocentric video recognition, leading to much higher efficiency. Our model is flexible to the target heavy video feature, as we demonstrate with multiple current leading egocentric video models [54, 16, 18, 17]. See Figure 1.

Importantly, EgoDistill offers a major upgrade in efficiency. Low-dimensional IMU and a few frames are much more efficient to process than a dense stack of frames. In practice, EgoDistill uses 200× fewer GFLOPs than the original video model. Furthermore, our approach is economical — the GoPro IMU sensor in our experiments (BOSCH BMI260) costs only $3 — and it is practical — most AR/VR devices and phones have in-built IMU sensors, making our method widely adoptable.

We experiment on the two largest egocentric action recognition datasets: Ego4D [26] and EPIC-Kitchens-100 [11]. We show that IMU coupled with an image offers better cross-modality knowledge distillation performance than images alone or images with audio. For a typical 50-minute egocentric video, EgoDistill reduces model inference time from 25 minutes to *36 seconds*. Moreover, with only 1-4 frames, our lightweight distillation model achieves a better accuracy-efficiency trade-off than state-of-the-art models for adaptively sampling video content [50, 65]. Notably, we surpass the accuracy of even these fast approaches by a large margin while requiring 4-8× less computation.

## 2 Related Work

**IMU for activity recognition.** Recent work explores using the IMU sensor on mobile devices for human activity recognition of actions like walking, jumping, or sitting [47, 59, 60, 3, 61]. Normally, these models use IMU sensors mounted on human body joints [60, 9, 42], waist-mounted [41] or in-pocket smartphones [33]. See [64] for a survey. Abundant work in video recognition explores ways to learn from RGB coupled with other modalities—audio [1, 22, 38], optical flow [57, 58, 19] or both [51, 36, 32]—but much fewer use IMU [48], and unlike our work, they focus on third-person video [68, 69, 13] and do not target model efficiency. Our idea is for IMU to help reconstruct expensive video features, rather than simply fuse IMU with RGB for multi-modal recognition.

**IMU for odometry.** Inertial odometry aims to estimate the position and orientation of the camera-wearer with readings from the IMU sensor. Traditionally, methods rely on IMU double integration [4] or enhancements thereof [40, 5, 35]. Recent data-driven methods automatically learn to perform inertial odometry with supervised [30, 70] or self-supervised learning [7], or combine IMU and

visual input for more robust estimates with visual-inertial odometry [20, 71]. While IMU can convey geometric ego-motion to our learned model, our goal is to produce efficient egocentric video features rather than to output odometry.

**Visual feature learning with IMU.** IMU is also used to learn better vision features [34, 14, 15, 63], e.g., to encourage image features that are equivariant with ego-motion [34], to predict an IMU-captured body part (leg, hand) [14, 15], or to predict video-IMU correspondence [63], for applications like action recognition [15, 63] and scene understanding [14, 34]. While these results reinforce that IMU can inject embodied motion into visual features, our idea to use head motion to infer pretrained video features for speedy video understanding is distinct.

**Efficient video recognition.** Being crucial for mobile applications, efficient video recognition has received increasing attention in recent years. Several studies focus on designing lightweight architectures [30, 17, 37, 72, 62] by reducing 3D CNN operations across *densely-sampled* frames. In contrast, our model focuses on inputs with *sparsely-sampled* frames. As we show in experiments, our method is compatible with different video architectures.

Another line of research achieves efficiency by adaptively selecting video content to process. Some reduce *temporal redundancy* by adaptively selecting which video clip [39], frames [49, 24], and/or feature channel [50] to process and which to skip, while others reduce *spatial redundancy*, selecting for each frame a smaller but important region to process [66, 67]. Other work dynamically selects tokens in video transformers among both the spatial and temporal dimensions [65]. Our idea is complementary: rather than dynamically subsample the available video content, we show how to *infer "full" video features for every clip* using static image(s) and motion data. Our results outperform state-of-the-art sampling models (cf. Sec. 4). In addition, we focus on egocentric video, where head motion is particularly meaningful for inferring unobserved visual content. To our knowledge, ours is the first technique specifically aimed at accelerating egocentric video processing.

**Multimodal distillation.** Knowledge distillation aims to transfer knowledge learned by an expensive model to a lightweight model [31]. Recent work explores multimodal distillation, e.g., transferring from a RGB model to a flow or depth model [23, 28], from a 3D model to a 2D model [45], or from a visual model to audio model [2, 21]. More related to our work, the ListenToLook model [22] incorporates both clip subsampling and video-to-audio distillation for fast activity recognition in third-person video. While we share a similar motivation, our contribution is distinct: we explore the relationship between the camera-wearer's head motion and RGB signals for egocentric video, and we introduce a novel IMU-based pretraining strategy. Our experiments show EgoDistill's advantage over ListenToLook in terms of the speed-accuracy tradeoff on egocentric video datasets. Furthermore, whereas [22] attempts only a single video model (R(2+1)D), we show our idea generalizes well to multiple video models, meaning it can be dropped in to benefit multiple popular frameworks.

## 3 Approach

We introduce EgoDistill, which uses sparsely-sampled frames and head motion from IMU to approximate the features of heavy video models for efficient egocentric video understanding. We first introduce the egocentric action recognition task (Sec. 3.1). Then, we introduce our pipeline (Sec. 3.2), our distillation model and training objective (Sec. 3.3), and our self-supervised IMU feature learning (Sec. 3.4). Figure 2 overviews our approach.

### 3.1 Egocentric action recognition

Given a fixed-length video clip $\mathcal{V} \in \mathbb{R}^{T \times H \times W \times 3}$ consisting of $T$ RGB frames of size $H \times W$ and a set of $C$ action classes, the task of action recognition is to output a score for each action class, representing its likelihood. Typically, this is done with a powerful but expensive video model $\Omega$, that directly operates on all the available frames to output the $C$ class logits $\Omega(\mathcal{V}) \in \mathbb{R}^C$. $\Omega$ is trained with standard classification loss:

$$\mathcal{L}_{\text{ACT}} = \sum_{\mathcal{V}_i} \mathcal{L}_{\text{CE}}(c_i, \sigma(\Omega(\mathcal{V}_i))), \tag{1}$$

where $\mathcal{V}_i$ is the $i$-th video clip in the dataset, $c_i$ is the corresponding ground-truth action label, $\sigma$ is the softmax function, and $\mathcal{L}_{\text{CE}}$ is cross-entropy loss. Popular video recognition models use clips that are typically ~2 seconds long [54, 16, 18]. For longer videos, scores are averaged across all clips it contains to infer the video action label.

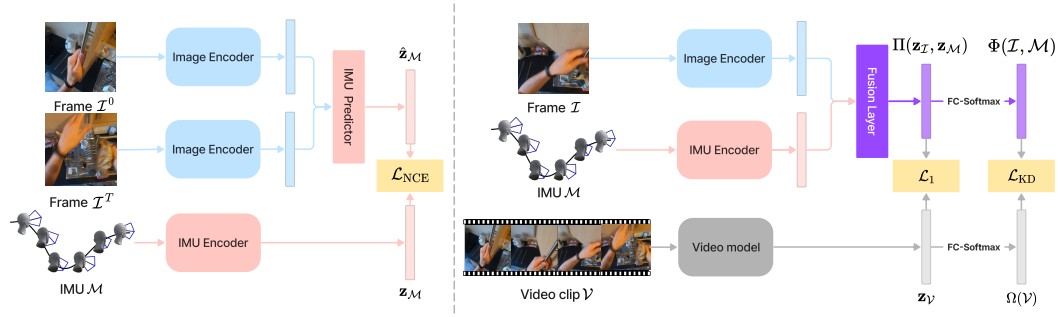

Figure 2: **EgoDistill architecture. Left: Self-supervised IMU feature learning.** Given start and end frames of a clip, we train the IMU encoder to anticipate visual changes. **Right: Video feature distillation with IMU.** Given image frame(s) and IMU, along with our pre-trained IMU encoder, our method trains a lightweight model with knowledge distillation to reconstruct the features from a heavier video model. When the input includes more than one image frame, the image encoder aggregates frame features temporally with a GRU.

## 3.2 Efficient video inference with head motion

Processing the video clip $\mathcal{V}$ for action recognition is computationally intensive; however, the computation cost can be modulated depending on how frames from the clip are used. On the one hand, *clip-based* models [18, 54, 16, 17] process most (or all) frames in a video clip $\mathcal{V}$ to achieve strong recognition performance, but come at a high computational cost. On the other hand, *frame-level* models [49, 24, 51, 67] only process one (or a small number) of frames from $\mathcal{V}$ and are more efficient, but suffer a drop in performance as a result. Our goal is to train a frame-based model that can approximate heavy clip-based model performance while maintaining high efficiency.

For this, we turn to head motion captured by IMU. Along with RGB frames, each video clip is paired with IMU measurements $\mathcal{M}$ that record the camera (head) motion during the video. Specifically, the IMU readings are composed of 6-dimensional accelerometer and gyroscope measurements in the $xyz$ axes, which encode strong temporal motion information about camera pose changes (both translation and rotation) across frames.

For short video clips, a set of sparsely sampled frames $\mathcal{I}$ often already captures most *appearance* information. Complementary to this, the IMU readings capture *camera motion* information (see below for discussion on scene motion). Moreover, IMU is very efficient to process due to its low dimensionality. By processing inputs from these two sources with a lightweight frame-based model, we can infer the semantic and dynamic features of a heavier clip-based video model.

Given $\mathcal{I}$ and $\mathcal{M}$, we train an efficient lightweight model $\Phi$ to approximate the output of video model $\Omega$. Specifically, we train our EgoDistill model $\Phi$ that achieves

$$\Phi(\mathcal{I}, \mathcal{M}) \approx \Omega(\mathcal{V}). \tag{2}$$

Such a lightweight model will be able to approximate the result of the heavy video model, while being much more efficient. Our approach is agnostic to the specific video model $\Omega$; in experiments, we demonstrate its versatility for MotionFormer [54], MViT [16], SlowFast [18] and X3D [17].

In practice, we uniformly sample $N$ frames[1] from $\mathcal{V}$ to obtain $\mathcal{I}$. We can achieve a trade-off between efficiency and performance by changing the number of frames $N$. In our experiments we use very low values of $N$ (1 to 4 frames). In the next section, we discuss how we train $\Phi$.

## 3.3 Video feature distillation with IMU

We address Equation 2 via knowledge distillation [31], where we transfer knowledge learned by the expensive teacher model $\Omega$ to a lightweight student model $\Phi$. Next we present the design of $\Phi$ and the training objectives, followed by our self-supervised IMU feature pretraining stage in Sec. 3.4.

---

[1]Other frame sampling heuristics (e.g., selecting from the start or center of the video) performed equivalently or worse than uniform sampling.

We design $\Phi$ to be a two-stream model. For a video clip and associated IMU signal $(\mathcal{I}, \mathcal{M})$, we extract image features $\mathbf{z}_{\mathcal{I}} = f_{\mathcal{I}}(\mathcal{I})$ and IMU features $\mathbf{z}_{\mathcal{M}} = f_{\mathcal{M}}(\mathcal{M})$ using lightweight feature encoders $f_{\mathcal{I}}, f_{\mathcal{M}}$ respectively. Then, we fuse $\mathbf{z}_{\mathcal{I}}$ and $\mathbf{z}_{\mathcal{M}}$ with a fusion network $\Pi$ to obtain the fused VisIMU feature $\mathbf{z}_{\phi} = \Pi(\mathbf{z}_{\mathcal{I}}, \mathbf{z}_{\mathcal{M}})$. Finally, a fully-connected layer uses the fused feature to predict class logits $\Phi(\mathcal{I}, \mathcal{M}) \in \mathbb{R}^C$. The fused feature $\mathbf{z}_{\phi}$ contains semantic information from the image frame coupled with complementary motion information from IMU, allowing us to accurately reconstruct the video clip feature. See Figure 2.

We train $\Phi$ with a combination of three losses, as follows. First, we train $\Phi$ to approximate the original video feature $\mathbf{z}_{\mathcal{V}}$ from the video model $\Omega$:

$$\mathcal{L}_1 = \sum_{(\mathbf{z}_{\mathcal{V}_i}, \mathbf{z}_{\phi_i})} \|\mathbf{z}_{\mathcal{V}_i} - \mathbf{z}_{\phi_i}\|_1. \tag{3}$$

This cross-modal loss encourages the fused feature $\mathbf{z}_{\phi}$ to match the video feature, i.e., the combined features from the different modalities should match in the feature space.

Training with $\mathcal{L}_1$ alone does not fully capture the classification output of $\Omega$. Therefore, we also train $\Phi$ with a knowledge distillation loss:

$$\mathcal{L}_{\text{KD}} = \sum_{(\mathcal{V}_i, \mathcal{I}_i, \mathcal{M}_i)} \mathcal{D}_{\text{KL}}(\sigma(\Omega(\mathcal{V}_i)/\tau), \sigma(\Phi(\mathcal{I}_i, \mathcal{M}_i)/\tau)), \tag{4}$$

where $(\mathcal{V}_i, \mathcal{I}_i, \mathcal{M}_i)$ represents the $i$-th clip in the dataset, $\mathcal{D}_{\text{KL}}$ measures KL-divergence between the class logits from the teacher model $\Omega$ and student model $\Phi$, and $\tau$ is a temperature parameter. Intuitively, $\mathcal{L}_{\text{KD}}$ casts the output of the video teacher model as a soft target for training the student model. In this way, the student model learns to better generalize by mimicking the output distribution of the heavy video model.

Finally, to further encourage the features to preserve elements useful for activity understanding, we also compute an action classification loss:

$$\mathcal{L}_{\text{GT}} = \sum_{(\mathcal{I}_i, \mathcal{M}_i)} \mathcal{L}_{\text{CE}}(c_i, \sigma(\Phi(\mathcal{I}_i, \mathcal{M}_i))), \tag{5}$$

where $c_i$ is the ground-truth action label, following Equation 1. The final training loss is a combination of these three loss functions:

$$\mathcal{L} = \alpha \mathcal{L}_{\text{KD}} + (1 - \alpha) \mathcal{L}_{\text{GT}} + \beta \mathcal{L}_1, \tag{6}$$

where $\alpha$ controls the balance between knowledge distillation and activity training [31], and $\beta$ controls the weight for feature space matching.

Critically, processing a few image frame(s) and the low-dimensional IMU readings is substantially faster than processing the entire video. Once trained, our model approximates the behavior of the source video model for recognition tasks, with the key benefit of efficient egocentric recognition.

What kind of motion does our model preserve? Video motion decomposes into *scene* motion (e.g., how the objects and the camera wearer's hands are moving on their own), and *camera* motion (i.e., how the camera wearer is moving their head). By itself, IMU would directly account only for camera motion, not scene motion. However, by learning to map from the RGB frame *and* IMU to the *full* video feature, we are able to encode predictable scene motions tied to scene content, e.g., how does hand and object movement in subsequent frames relate to the camera wearer's head motion. Moreover, our model is applied to relatively short clips (1-2 seconds) in sequence, which means the appearance content is regularly refreshed as we slide down to process the longer video.

### 3.4 Self-supervised IMU feature learning

The success of EgoDistill depends on how well the IMU feature encoder $f_{\mathcal{M}}$ extracts useful camera motion information and associates it with the visual appearance change in the video clip. In this way EgoDistill can learn to anticipate unseen visual changes in the video with $\mathcal{I}$ and $\mathcal{M}$. We design a self-supervised pretraining task to initialize the weights of $f_{\mathcal{M}}$ to achieve this.

Specifically, for each clip $\mathcal{V}$, we obtain its first and last frames $(\mathcal{I}^0, \mathcal{I}^T)$ as well as the IMU $\mathcal{M}$. We first extract visual features $\mathbf{z}_{\mathcal{I}}^0, \mathbf{z}_{\mathcal{I}}^T$ and IMU feature $\mathbf{z}_{\mathcal{M}}$ with feature extractors $f_{\mathcal{I}}$ and $f_{\mathcal{M}}$

mentioned above. Then, we train a feature predictor $h$ to predict the IMU feature $\hat{\mathbf{z}}_{\mathcal{M}} = h(\mathbf{z}_{\mathcal{I}}^0, \mathbf{z}_{\mathcal{I}}^T)$. By connecting $\hat{\mathbf{z}}_{\mathcal{M}}$—a function of image features only—with $\mathbf{z}_{\mathcal{M}}$, we encourage $f_{\mathcal{M}}$ to extract useful camera motion features specifically associated with the visual appearance changes. Note that those appearance changes may include scene motion. Therefore, we include an $\mathcal{L}_1$ loss to train $f_{\mathcal{M}}$, which encourages $f_{\mathcal{M}}$ to extract motion features accounting for scene motion in the full video.

We train $f_{\mathcal{M}}$, $h$, and the fusion network $\Pi$ using $\mathcal{L}_1$ and NCE loss [29]: $\mathcal{L}_{\text{pretrain}} = \mathcal{L}_{\text{NCE}} + \mathcal{L}_1$, where

$$\mathcal{L}_{\text{NCE}} = \sum_i -\log \frac{\text{sim}(\hat{\mathbf{z}}_{\mathcal{M}_i}, \mathbf{z}_{\mathcal{M}_i})}{\sum_j \text{sim}(\hat{\mathbf{z}}_{\mathcal{M}_i}, \mathbf{z}_{\mathcal{M}_j})}. \tag{7}$$

We sample negative examples $\mathbf{z}_{\mathcal{M}_j}$ from other instances in the same mini-batch for $j \neq i$, and $\text{sim}(q, k) = \exp(\frac{q \cdot k}{|q||k|} \frac{1}{\tau'})$ with temperature $\tau' = 0.1^2$.

To summarize, prior to the main training stage of Equation 6, we pretrain the IMU feature extractor $f_{\mathcal{M}}$ and fusion network $\Pi$. As we will show below, both pretraining losses result in IMU features that are consistent with visual changes and lead to better finetuning performance.

## 4  Experiments

We evaluate our approach for resource-efficient action recognition.

### 4.1  Experimental setup

**Datasets.** We experiment on two large-scale egocentric action recognition datasets. (1) **Ego4D** [26] contains 3,670 hours of egocentric videos of people performing diverse tasks (from cooking to farming) across the globe. As action recognition is not part of the original Ego4D benchmark, we construct this task with annotations from the Hands+Objects temporal localization benchmark [26] (see Supp. for details). We include clips with paired IMU and audio[3], and consider classes with at least 2 labeled instances. This results in a 94-class action recognition dataset with 8.5k training videos and 3.6k evaluation videos. (2) **EPIC-Kitchens** [11] contains 100 hours of egocentric videos capturing daily activities in kitchen environments. We use annotations from the action recognition benchmark. Similar to Ego4D, we select videos that have paired IMU and audio data, and split the resulting data by camera-wearer. This results in a 62-class action dataset with 29k training videos and 6.2k evaluation videos. For both datasets, we use "verb" labels as the target for action recognition as they are well aligned to activity motions.

**Evaluation metrics.** To measure action recognition performance, we report the per-video top-1 accuracy on the validation set. We densely sample clips from each video and average their predictions to compute accuracy. To benchmark efficiency, we measure computational cost with FLOPs (floating-point operations) during inference.

**Implementation details.** In our main experiments, we use MotionFormer [54] as the video teacher model $\Omega$ due to its strong performance for egocentric video. For EPIC-Kitchens, we use the authors' provided checkpoint. For Ego4D, we finetune the above model for 50 epochs with $1e^{-4}$ learning rate and 64 batch size on the training set. We use 16-frame input with sample rate 4. For the student model $\Phi$, we use a ResNet-18 as the image backbone $f_{\mathcal{I}}$ and a 1D Dilated CNN [6] for the IMU backbone $f_{\mathcal{M}}$. The feature fusion module $\Pi$ uses a concatenation operation following a two-layer fully-connected layer with hidden dimension 1024. For each video clip, the input image(s) is resized to $224 \times 224$, and the IMU is a $422 \times 6$ matrix (around 2 seconds with 198Hz frequency), representing the accelerometer and gyroscope readings along the $xyz$ axes. For the image input, we uniformly sample $N$ frames from the video clip. If $N > 1$, we use $f_{\mathcal{I}}$ to sequentially generate features for each frame and aggregate them with a GRU module [10]. For both datasets, we first pretrain the model with the self-supervised objective (Sec. 3.4) for 50 epochs with AdamW [46] using batch size 64 and learning rate $1e^{-4}$. Then, we finetune all the models with the same setting (Equation 6). We set $\alpha = 0.95$ and $\beta = 1.0$ based on validation data. For Ego4D, we set $\tau = 10.0$ and train the model for 150 epochs. For EPIC-Kitchens, we set $\tau = 1.0$ and train for 50 epochs.

---

[2]We keep the ImageNet-pretrained $f_{\mathcal{I}}$ model frozen, as finetuning it leads to mode collapse.
[3]We require audio to compare with the audio-based baseline [22].

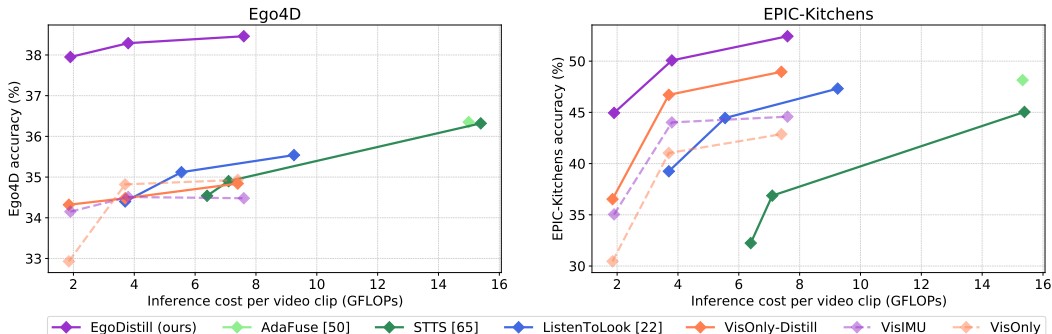

Figure 3: **Accuracy vs. efficiency for action recognition on Ego4D (left) and EPIC-Kitchens (right).** EgoDistill outperforms state-of-the-art efficient video recognition methods that adaptively sample video content, while using $4\times$ to $8\times$ fewer GFLOPs.

## 4.2 Baselines

We compare with the following methods: (1) **AdaFuse** [50] trains a lightweight policy network to adaptively compute (or skip) feature map channels for each frame during inference. We use the AdaFuse$_{\text{R50}}^{\text{TSN}}$ model with the provided hyper-parameters. (2) **STTS** [65] trains a module to rank spatio-temporal tokens from videos in a transformer-based model, and selects only the top-K tokens to speed up inference. We use a MViT-B16 backbone [17]. (3) **ListenToLook** [22] uses the audio-based distillation module from [22] following the same audio processing and model architecture.

These methods represent recent advances in efficient video recognition models. AdaFuse represents state-of-the-art approaches that achieve efficiency by reducing temporal redundancy in CNN models. STTS is one of the most recent approaches that efficiently reduces both spatial and temporal redundancy in ViT models, which achieves the state-of-the-art on Kinectics-400 [8]. ListenToLook also relies on distillation, but using audio rather than head motion. For each model we generate multiple versions with different computation budgets to plot accuracy vs. GFLOPs. We train all AdaFuse and STTS models with 4 input frames to align with the maximum frames used by our model. For AdaFuse, we use the only provided hyper-parameter in the paper.[4] For STTS, we use three provided variants: $T_{0.5}^0$-$S_{0.7}^4$, $T_{0.8}^0$-$S_{0.9}^4$ and the full model without token selection. For ListenToLook we adopt the same efficiency-accuracy trade-off as our method, i.e., varying the number of input frames.

We also test variants of our method: (1) **VisOnly-Distill** is our model without the IMU branch and fusion layer, but trained with the same loss function. This model reveals the importance of IMU for distillation. (2) **VisIMU** is our model trained with only $\mathcal{L}_{\text{GT}}$ in Equation 5. It shows the effectiveness of distillation from the video model compared with directly training the features with action labels. (3) **VisOnly** is an image-only model trained with $\mathcal{L}_{\text{GT}}$, which serves as the baseline.

## 4.3 Main Results

**Importance of IMU-guided distillation.** Figure 3 shows the accuracy vs. efficiency curves. Methods towards the top-left of the plot represent those with both high accuracy and efficiency. Our method achieves good accuracy with low computational cost. Specifically, on EPIC-Kitchens, when $N = 1$, EgoDistill improves over VisOnly-Distill by $8.4\%$ with only a small increase in computation. This shows the effectiveness of IMU for reconstructing egocentric video features. Compared to VisIMU, EgoDistill improves by $9.9\%$, showing the effectiveness of knowledge distillation from the video model. Importantly, this reveals that EgoDistill does not simply benefit from the extra IMU context; our idea to approximate video features is necessary for best results. We see similar results on Ego4D.

**Comparison with the state of the art.** Figure 3 also shows that EgoDistill achieves better accuracy with less computation than existing efficient video recognition models AdaFuse [50], STTS [65], and ListenToLook [22]. With $N = 4$ frames, EgoDistill surpasses STTS by $7.4\%$ and AdaFuse by $4.2\%$ on EPIC-Kitchens, with $2\times$ fewer GFLOPs, and surpasses both methods by $2.1\%$ on Ego4D. When we use fewer frames, EgoDistill can still outperform STTS and AdaFuse using $4\times$ to $8\times$ fewer

---

[4]Modifying hyper-parameters to control the accuracy-efficiency trade-off results in unstable training and unreliable performance.

| $\mathcal{L}_{KD}$ | $\mathcal{L}_1$ | $\mathcal{L}_{GT}$ | $\mathcal{L}_1$-pretrain | $\mathcal{L}_{NCE}$-pretrain | Ego4D | EPIC-Kitchens |
|---|---|---|---|---|---|---|
| | | ✓ | | | 34.15 | 35.04 |
| | ✓ | ✓ | ✓ | ✓ | 35.51 | 39.33 |
| ✓ | | ✓ | ✓ | ✓ | 37.71 | 42.20 |
| ✓ | ✓ | | ✓ | ✓ | 37.46 | 43.17 |
| ✓ | ✓ | ✓ | | | 36.99 | 41.21 |
| ✓ | ✓ | ✓ | | ✓ | 37.26 | 42.30 |
| ✓ | ✓ | ✓ | ✓ | | 37.49 | 43.51 |
| ✓ | ✓ | ✓ | ✓ | ✓ | **37.95** | **44.95** |

Table 1: **Ablation study of EgoDistill's model components.** Accuracy with $N = 1$.

| Source Model | Ego4D | | | EPIC-Kitchens | | |
|---|---|---|---|---|---|---|
| | Video | EgoDistill | VisOnly-D | Video | EgoDistill | VisOnly-D |
| MFormer [54] | 46.38 | **37.95** | 34.32 | 77.28 | **44.95** | 37.20 |
| MViT [16] | 40.32 | **36.46** | 33.40 | 53.38 | **36.90** | 31.22 |
| SlowFast [18] | 40.52 | **33.29** | 33.04 | 58.34 | **39.42** | 33.47 |
| X3D [17] | 37.56 | **33.57** | 32.90 | 52.28 | **36.34** | 31.71 |

Table 2: **Versatility to model architectures.** EgoDistill outperforms the baselines for multiple common architectures, showing the generality of our idea. "Video" refers to the more expensive source model, essentially the upper bound for accuracy. We show the model accuracy under $N = 1$.

GFLOPs. In addition, EgoDistill surpasses ListenToLook by $7.4\%$ and $2.9\%$ on EPIC-Kitchens and Ego4D respectively, which suggests that head motion is more informative than audio for feature reconstruction in egocentric video.

## 4.4 Analysis

**Model component ablations.** Table 1 ablates different design choices in our model, setting $N = 1$ for all experiments. We observe that training EgoDistill without $\mathcal{L}_1$, $\mathcal{L}_{KD}$ or $\mathcal{L}_{GT}$ deteriorates performance. Specifically, training without $\mathcal{L}_{KD}$ leads to the largest performance drop, which indicates that knowledge distillation is an essential component in our approach. Training without $\mathcal{L}_1$ also degrades performance, which shows the importance of our idea to align features from the different modalities. Further, our self-supervised pretraining stage is very effective at training the IMU extractor to encode useful motion information that is consistent with visual feature change. We find an alternative IMU extractor pretraining strategy of supervised action recognition task is inferior; we outperform pretaining with action recognition by $3.74\%$ on EPIC-Kitchens and $0.96\%$ on Ego4D. Finally, we compare with a model that simply does multi-modal recognition with IMU (top row). The strong contrast here indicates the importance of our idea to use IMU to predict video model features, as opposed to simply adding IMU as an additional input modality.

**Impact of teacher video model architecture.** In our main experiments we use MotionFormer [54] as the teacher video model due to its strong performance on egocentric video tasks. To emphasize the generality of our idea, we show the performance of EgoDistill with other video teacher architectures in Table 2. Similar to the MotionFormer model, we train these models on each of the labeled datasets, and then train our model using the resulting video models as the teacher. As expected, better video teacher models lead to better student model performance. More importantly, we observe consistent improvement by EgoDistill over the VisOnly-Distill baseline on both datasets and with different video teacher models, highlighting our idea's generality and versatility.

**Efficiency analysis.** To compare the efficiency of different models, aside from GFLOPs, we also compare their inference run-time and number of parameters. For run-time, we record the time spent to infer a single video clip's label with a single A40 GPU, and take the average time over the full validation datasets of Ego4D and EPIC-Kitchens with batch-size of 32. Table 3 shows the results. EgoDistill runs much faster than the other methods. Notably, it reduces the GFLOPs of MotionFormer by nearly $200\times$. Furthermore, it runs $6.5\times$ faster than STTS [65] while achieving $4.4\%$ higher accuracy on EPIC-Kitchens.

**Effect of fusing IMU data.** EgoDistill achieves superior performance by combing IMU information when distilling features from video models. Here we explore how helpful it is to combine IMU

| | GFLOPs | Runtime (ms) | Parameters (M) |
|---|---|---|---|
| Video [54] | 369.51 | 10.70 | 108.91 |
| AdaFuse [50] | 15.20 | 2.04 | 38.85 |
| STTS [65] | 7.19 | 1.63 | 36.63 |
| ListenToLook [22] | 3.10 | 0.43 | 25.53 |
| EgoDistill | **1.91** | **0.25** | **20.56** |

Table 3: **Efficiency analysis.** Our approach is the most efficient. "Video" refers to the original (full-clip) feature. Lower is better.

| Method | Ego4D | | EPIC-Kitchens | |
|---|---|---|---|---|
| | w/ IMU | w/o IMU | w/ IMU | w/o IMU |
| STTS [65] | 34.93 | 34.54 | 38.38 | 32.24 |
| EgoDistill | **37.95** | 34.32 | **44.95** | 37.20 |

Table 4: **Effect of combining IMU with vision as input.** In both EgoDistill and STTS, our method is able to combine vision and IMU information more effectively.

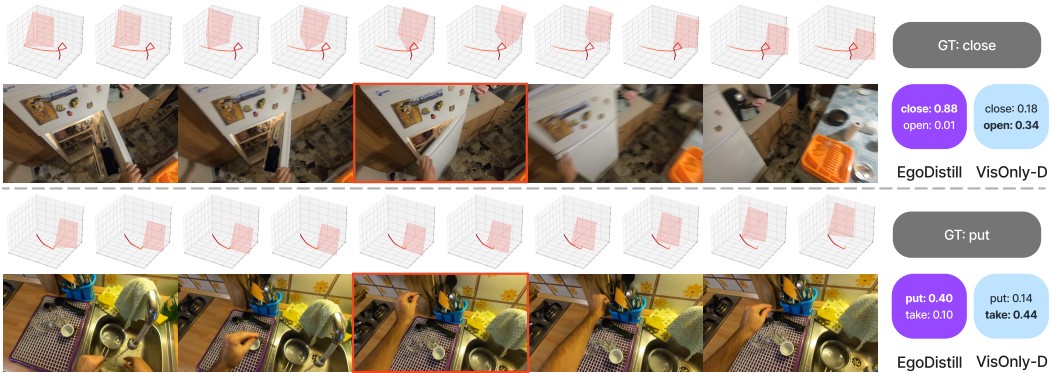

Figure 4: **Anticipating scene motion with EgoDistill.** For each clip, we show the head motion and video frames. Note, only the center frame (red border) is observed by the model. Action classification scores are shown on the right. EgoDistill can successfully anticipate scene motion and disambiguate the action semantics in the input frame. For example, in the top center frame, the image alone cannot reveal if the door is being opened or closed, whereas our feature, learned with head motion, recovers correlations with the *scene motion* (i.e., hand motion and door motion) to disambiguate "close" from "open". A similar effect for "put" vs. "take" is seen in the second example.

information with the most recent efficient video understanding method, *i.e.*, STTS [65].[5] To this end, for STTS, we add an IMU encoding branch identical to the one used in EgoDistill, and concatenate the IMU feature with video features before the last FC layer. We train both the video and IMU branches simultaneously for the action recognition task. Results in Table 4 show that combining IMU features indeed helps STTS, yet EgoDistill's performance is significantly better than STTS's. These results again indicate that our advantage is not simply access to IMU; rather it is the proposed way we leverage vision and IMU. The reason that EgoDistill is able to outperform STTS+IMU is mainly that EgoDistill learns to jointly distill information from a video feature with image and IMU inputs. This distillation process will encourage the model to extract the interplay between static semantic information in the image and the dynamic information in IMU, as it is trained to reconstruct the video model feature. On the other hand, simple concatenation of the IMU feature to the image feature for action classification will not exploit the relationship between the image and IMU—it lacks the training signal to target the dynamic video clip feature—leading to inferior performance.

**Effect of using partial IMU**. Is the net displacement of the camera what matters, or does the motion in between give cues? To answer this, we experiment with feeding EgoDistill with partial IMU readings of the full video clip. Specifically, we train and evaluate our model with only the first K% of IMU readings of the full clip and pad the rest with 0. We show the results in Table 5. This ablation study shows that more complete IMU readings yield the best results, while using 25% and 50% IMU data achieves the majority of the gain over the VisOnly-D (no IMU) baseline.

**Effect of number of input frames**. To investigate the effect of using different numbers of input frames, we conduct an ablation study with $N$. Specifically, we use three settings: $N = 1$ (use the center frame), $N = 2$ (use the start frame and end frame), $N = 4$ (uniformly sample 4 frames). We compare EgoDistill and VisOnly-D under different settings in Table 6. We observe that only using the overall displacement between the end frame and the starting frame ($N = 2$) leads to better

---

[5]We tried to augment AdaFuse [50] the same way, but adding IMU to its codebase gave unreliable results.

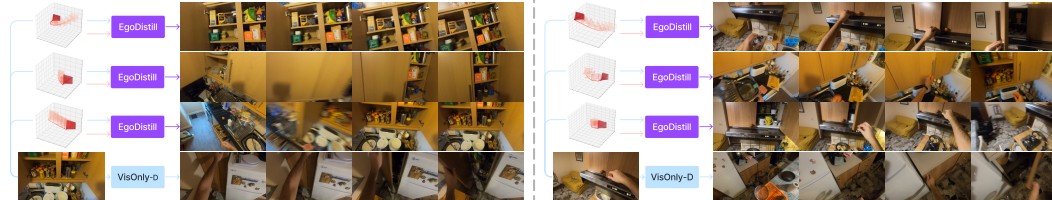

Figure 5: **Retrieving video clips with EgoDistill.** Given a query frame (bottom left) and a paired IMU segment (red camera frustums) , we retrieve the nearest clip in the video dataset according to EgoDistill and visualize its (unobserved) frames (strip to the right). Compared to VisOnly-Distill, which outputs a single feature for a given input frame (bottom row), EgoDistill outputs a distinct feature by conditioning on IMU, showing its ability to preserve both semantic and motion during reconstruction. For instance, in the top-right example, EgoDistill retains the cabinet interaction semantics in the frame as well as the upward camera-motion in the IMU. Zoom in to view best.

| Input IMU | Ego4D | EPIC-Kitchens |
|---|---|---|
| 100% (EgoDistill) | 37.95 | 44.95 |
| 75% | 37.88 | 43.43 |
| 50% | 37.65 | 43.69 |
| 25% | 37.21 | 42.68 |
| 10% | 36.60 | 41.45 |
| 0% (VisOnly-D) | 34.32 | 37.20 |

| Input Frame | Ego4D | | EPIC-Kitchens | |
|---|---|---|---|---|
| | EgoDistill | VisOnly-D | EgoDistill | VisOnly-D |
| N=1 (center) | 37.95 | 34.32 | 44.95 | 37.2 |
| N=2 (first and last) | 38.29 | 34.48 | 50.07 | 46.71 |
| N=4 (uniform) | 38.46 | 34.84 | 52.43 | 48.96 |

Table 5: **Effect of partial IMU.** We observe using partial IMU input already achieves significant gain.

Table 6: **Effect of number of input frames.** Using more input frames leads to larger improvement of EgoDistill over VisOnly-D baseline.

performance by VisOnly-D than using only one frame ($N = 1$). However, the motion trajectory still helps EgoDistill obtain significantly better performance in either case. This result indicates the information in the motion trajectory captured by IMU. Note that while $N = 2, 4$ leads to better performance, it requires higher computational cost.

**What do EgoDistill features capture?** To explore this, we pair a single input frame with different IMU clips as inputs to EgoDistill, then retrieve the nearest video clip for each resulting anticipated video feature. We also compare with the VisOnly-Distill result. Figure 5 illustrates this. We see that EgoDistill outputs video features that all involve interaction with the cabinet (right panel), and is able to use different IMU inputs to retrieve different video clips that show consistent camera motion. In contrast, different input IMUs lead to corresponding camera motion in the frames. The result shows that EgoDistill approximates video features that capture both semantic and motion information, whereas VisOnly-Distill only retains the semantic context to retrieve a single clip.

**Is there evidence EgoDistill captures scene motion?** Figure 4 shows how our features learned with *head* motion can nonetheless expose certain scene motion cues. EgoDistill improves the accuracy over VisOnly-Distill on ambiguous categories (like *close* and *put*) by a large margin (20.3% and 10.4% on EPIC-Kitchens, 8.5% and 3.9% on Ego4D). See caption and **Supp. video** for more details.

**Limitations of EgoDistill**. Although EgoDistill is useful for many action categories in egocentric videos, for some other classes it is less effective. We observe that video clips that EgoDistill does not perform well tend to contain little head motion —in which case IMU is redundant to the RGB frame— or drastic head motion that is weakly correlated with the camera wearer's activity and introduces blur to the frame. Additionally, EgoDistill might not be useful when we want to focus on the "noun" part of the action (e.g., differentiating watching TV and using the cell phone), as IMU information does not contain information about the object semantics in the scene.

## 5   Conclusion

We present EgoDistill, the first model to explore egocentric video feature approximation for fast recognition. Experiments on action recognition on Ego4D and EPIC-Kitchens demonstrate that our model achieves a good balance between accuracy and efficiency, outperforming state-of-the-art efficient video understanding methods. Our approach has great potential to accelerate video understanding for egocentric videos using a data stream that is already ubiquitous in egocentric cameras. In the future, we plan to investigate how to use head motion for long-term human activity understanding with room context and visual correspondence learning for multi-view videos.

**Acknowledgements** We thank the UT Austin vision group for helpful discussions. We also thank the anonymous reviewers for their insightful suggestions. UT Austin is supported in part by IFML NSF AI Institute. KG is paid as a research scientist at Meta.

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
