# EgoDistill: Egocentric Head Motion Distillation for Efficient Video Understanding

**Shuhan Tan**[1]**, Tushar Nagarajan**[2]**, Kristen Grauman**[1,2]
[1]University of Texas at Austin, [2]FAIR, Meta

The supplementary materials of this work consist of:

A. Supplementary video.

B. Dataset details.

C. Implementation details.

D. AdditionalOther analysis of our model.

## A   Supplementary Video

We include our supplementary video on our project page `https://vision.cs.utexas.edu/projects/egodistill/`. In our supplementary video, we have a brief introduction of our work. More importantly, we show animated videos of Best and Worse reconstructed clips (Figure A2), and Anticipating scene motion with EgoDistill (Figure 6 in our main paper).

Animated version of these figures better show head motion and video dynamics. We recommend viewing the supplementary video for better understanding of our method and results.

## B   Dataset Details.

We use two datasets in our experiments: Ego4D (4) and EPIC-Kitchens-100 (2). In this section we describe more details about how we create our training and evaluation data.

1. **Ego4D** (4) contains 3,670 hours of egocentric videos of people performing diverse tasks (from cooking to farming) across the globe. As action recognition is not part of the original Ego4D benchmark, we construct this task with annotations from the Hands+Objects temporal localization benchmark (4). Specifically, for each hand-objects interaction temporal annotation, we take the video clip between the pre-frame and post-frame of the annotation as input, and use the annotated verb for this interaction as label.

   We include clips with paired IMU and audio, and consider classes with at least 2 labeled instances, resulting in 94 action categories with 12.1k videos in total. In average, each clip has 2.2 second duration. Then, we randomly split data from each category into training and evaluation sets with 70%:30% ratio. Finally, we obtain a 94-class action recognition dataset with 8.5k training videos and 3.6k evaluation videos.

2. **EPIC-Kitchens** (2) contains 100 hours of egocentric videos capturing daily activities in kitchen environments. We use annotations from the action recognition benchmark in our experiment.

   We select videos that have paired IMU and audio data, and split the resulting data by camera-wearer, ensuring non-overlapping splits following the original benchmark setting. Specifically, we take videos captured by camera-wearer id starting with P30, P35, P37 as evaluation videos and use all the remaining videos as training videos. This results in a 62-class action dataset with 29k training videos and 6.2k evaluation videos.

37th Conference on Neural Information Processing Systems (NeurIPS 2023).

|          | Ego4D | EPIC-Kitchens |
|----------|-------|---------------|
| uniform  | 38.46 | 52.43         |
| random   | 36.85 | 48.48         |
| first    | 38.68 | 46.40         |
| last     | 35.46 | 41.72         |
| center   | 37.04 | 44.85         |

Table A1: **Effect of frame selection.** We compare the accuracy of using different frame selection heuristics for EgoDistill when $N = 4$. We observe that Uniform on average achieves better results.

## C Implementation Details.

**IMU input processing.** For each input clip, IMU input is a $422 \times 6$ matrix (around 2 seconds with 198Hz frequency), representing the accelerometer and gyroscope readings along the $xyz$ axes. We observe that the raw IMU input has significant drifting and bias issues. This induces inconsistent correspondence between camera motion and IMU reading across different clips and videos. Therefore, for IMU reading of each clip, on each dimension we separately subtract raw readings by the mean values on the corresponding dimension. This operation normalizes IMU readings in each dimension to have zero average value. In this way, our model can only focus on the temporal motion patterns in each clip.

**Audio input processing.** For ListenToLook (3), we process the audio input in the same way mentioned in the paper. Specifically, we subsample the audio at 16kHZ, and compute STFT using Hann window size of 400 and hop length of 160. Please refer to (3) for more details.

**Model architecture.** For the image backbone, we use the ImageNet-pretrained ResNet-18 model. For the IMU backbone, we use a 5-layer 1D Dilated CNN, as found effective for IMU data processing (1). We use the same network setting (kernel dimension, dilation gap and channel dimension) as in prior work (1). The feature fusion model consists of a concatenation operation following two fully-connected layers with hidden dimension of 1024. Each layer except for the output layer is followed by a ReLU activation. The output dimension is the same as the teacher video model's feature dimension (768 in the case of MotionFormer). When $N > 1$, we use a one-layer GRU module to aggregate extracted features for each frame. We use a single-directioal GRU with hidden dimension of 512.

**Model training.** We train our models in two stages. In the self-supervised IMU feature learning stage, we train random initialized IMU encoder $f_{\mathcal{M}}$, IMU predictor $h$ and the fusion network $\Pi$ with $\mathcal{L}_{\text{NCE}}$. Here the image encoder $f_{\mathcal{I}}$ is a fixed ImageNet pretrained model. On both datasets, we train the model for 50 epochs with AdamW and batch size 64. The initial training rate is $1e^{-4}$. We decay the training rate by $0.1$ at epoch 30 and epoch 40. In the second video feature distillation stage, we initialize the model with parameters obtained in the last stage and finetune. On both datasets, we use AdamW with batch size 64 and initial learning rate $1e^{-4}$. On Ego4D, we train for 150 epochs. We decay the training rate by $0.1$ at epoch 90 and epoch 120. On EPIC-Kitchens, we train for 50 epochs. We decay the training rate by $0.1$ at epoch 30 and epoch 40.

## D Analysis.

**Effect of frame selection.** In Section 3.2, we mentioned that we use uniform sampling to obtain the $N$ frames from each video clip. In this section, we compare the performance of our work under uniform sampling with other heuristics. Specifically, we compare with random sampling, the first $N$ frames, the last $N$ frames and the center $N$ frames. We show the results in Table A1 under $N = 4$. These results indicate that uniform sampling leads to the best performance on average. Intuitively, uniform sampling on average leads to a broader coverage of both semantic contexts as well as scene motion.

**Why we set $N$ to be small.** In our experiments, we set $N$ to be 1 to 4. Using larger $N$ (*e.g.,* 8 or 16) with densely sampled frames could lead to better results of all the methods with more computational cost. Efficient video understanding methods could benefit more as they have better

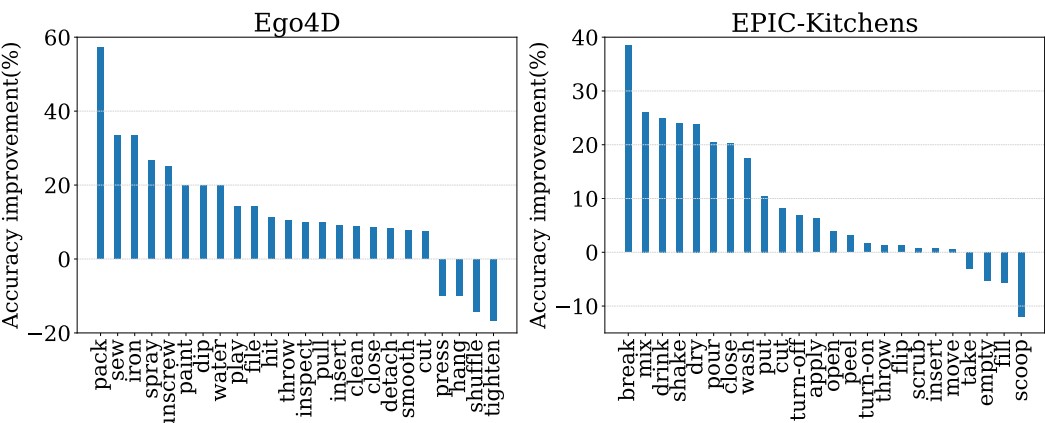

Figure A1: **Per-class accuracy improvement** over VisOnly-Distill. Best and worst performing classes are shown.

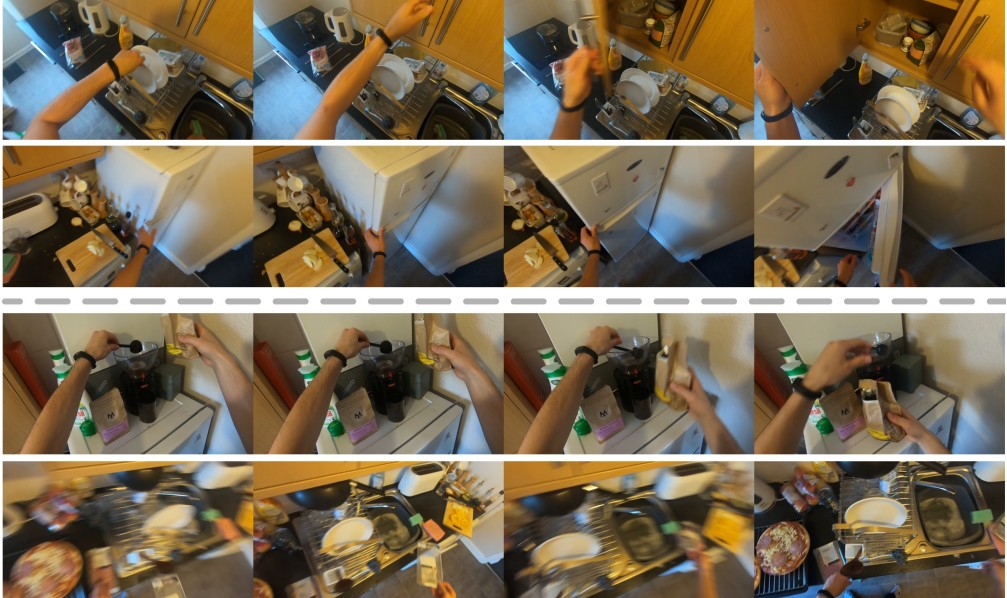

Figure A2: **Best (top) and worst (bottom) reconstructed videos.**

temporal aggregation mechanisms given densely-sampled frames. However, the core purpose of our model is to deal with cases where we only use a few number of samples. Therefore, our model is not comparable to video clip models under dense-frame setting. Furthermore, setting $N$ to be a small number is very important in many applications. As loading more image frames takes additional time and memory, applications with streaming videos or low-resource AR/VR devices will benefit from loading only a few frames.

**Where does our model work best/worst?** In Figure 3 of our main paper, we saw that using IMU leads to an overall performance improvement on action recognition, indicating better video feature prediction capability. In this section, we further Next, we explore what kinds of clips are better reconstructed using EgoDistill. Figure A1 shows the improvement of EgoDistill over the VisOnly-Distill model on Ego4D and EPIC-Kitchens split by action class. We observe that IMU is more useful for actions with predictable head motion (e.g., *break*, *cut*, *close*), and is less helpful for actions where head motion may be small or unrelated (e.g., *empty*, *fill*, *press*).

Figure A2 shows clip examples whose video features are best and worst reconstructed. We observe that the best reconstructed clips (top) contain moderate head motion that is predictive of scene

motion and action semantics. For example, the camera wearer's head moves slightly backwards while opening the cabinet. On the other hand, more poorly reconstructed clips tend to contain little head motion (third row)—in which case IMU is redundant to the RGB frame—or drastic head motion that is weakly correlated with the camera wearer's activity and introduces blur to the frame (last row).