# OpenReview forum: "EgoDistill: Egocentric Head Motion Distillation for Efficient Video Understanding"
_NeurIPS.cc/2023/Conference — NeurIPS 2023 poster_

### Official Review · Reviewer_F46j · 2023-07-01

**Soundness:** 1 poor
**Presentation:** 3 good
**Contribution:** 2 fair
**Rating:** 4
**Confidence:** 3

**Summary:**

To improve the efficiency of ego-centric video representation learning, this paper presents a distillation-based approach that learns to reconstruct heavy egocentric video clip features from the combination of sparsely-sampled video frames and the camera motion from IMU.

**Strengths:**

1. The paper is well-written and the easy to follow.

2. Improving the efficiency of ego-centric video representation learning is indeed valuable.

**Weaknesses:**

1. As mentioned in the paper, the dynamics in a video include both the movement of the scene and the camera (Line 34-35). I agree that IMU could provide some cues for the motion of the camera, however, I'm afraid it is difficult to deduce the motion of the scenes, which is more important for the video understanding (although some analysis is presented in Figure 6, it is still contradictory to the intuition).

2. From Table 1, we can see that the performance improvements mainly come from L_{KD}, other components like L1, L1-pretrain, and L1-NCE, bring subtle improvements.

3. It could be better to compare the model performance in Table 4 (which is mismarked as Figure 4).

**Questions:**

Please refer to the weakness.

**Limitations:**

The authors have discussed the potential limitations in the paper.

---

> ### Author Rebuttal · Authors · 2023-08-10
>
> Thank you for your review and constructive comments! We’ll address your questions point-by-point:
>
> ---
>
>
> > 1. As mentioned in the paper, the dynamics in a video include both the movement of the scene and the camera (Line 34-35). I agree that IMU could provide some cues for the motion of the camera, however, I'm afraid it is difficult to deduce the motion of the scenes, which is more important for the video understanding (although some analysis is presented in Figure 6, it is still contradictory to the intuition).
>
>
> In L182-189, we discussed why we think EgoDistill could learn scene motion cues from IMU. In short, we believe this is because EgoDistill learns to map from the RGB frame (**scene context**) **and** IMU (**head motion**) to full video feature (scene motion + head motion).  In this way, it learns to encode predictable scene motions tied to scene context. Please refer to the paper for more details.
>
> In addition to the discussion in the paper, we conduct two more additional analyses:
>
> (Please refer to the attached **PDF** file in the overall author rebuttal for the mentioned figures.)
>
>
> **To understand how information in IMU contributes to the performance boost, we analyze when IMU is useful and its limitations:**
>
> We saw in our paper that using IMU leads to an overall performance improvement in action recognition, indicating better video feature prediction capability (Lines 265-272 and Figure 3).
>
> Here, we explore what kinds of clips are better reconstructed using EgoDistill. Figure.A1 in the uploaded pdf file shows the improvement of EgoDistill over the VisOnly-Distill model on Ego4D and EPIC-Kitchens split by action class.  We observe that IMU is more useful for actions with predictable head motion (e.g., *break*, *cut*, *close*), and is less helpful for actions where head motion may be small or unrelated (e.g., *empty*, *fill*, *press*).
>
> Figure.A2 in the uploaded pdf file shows clip examples whose video features are best and worst reconstructed.  We observe that the best-reconstructed clips (top) contain moderate head motion that is predictive of scene motion and action semantics. For example, the camera wearer's head moves slightly backward while opening the cabinet.  On the other hand, more poorly reconstructed clips tend to contain little head motion (third row)---in which case IMU is redundant to the RGB frame---or drastic head motion that is weakly correlated with the camera wearer's activity and introduces blur to the frame (last row).
>
>
> **To understand what information does EgoDistill extract from IMU data, we conduct an experiment to explore the feature extracted from EgoDistill:**
>
> To explore what our model learns, we pair a single input frame with different IMU clips as inputs to EgoDistill, then retrieve the nearest video clip for each resulting anticipated video feature.  We also compare with VisOnly-Distill result.
>
> Figure.A3 in the uploaded pdf illustrates this.  We see that EgoDistill outputs video features that all involve interaction with the cabinet (right panel), and is able to use different IMU inputs to retrieve different video clips that show consistent camera motion. In contrast, different input IMUs lead to corresponding camera motion in the frames.  The result shows that EgoDistill is able to approximate video features that capture both semantic and motion information, VisOnly-Distill only retains the semantic context to retrieve a single clip. These results indicate that EgoDistill is able to approximate video features that capture both semantic and motion information.
>
> We thank you for pointing out this question and will add this additional analysis in our final version.
>
> ---
>
>
> > 2. From Table 1, we can see that the performance improvements mainly come from L_{KD}, other components like L1, L1-pretrain, and L1-NCE, bring subtle improvements.
>
> Actually, the other components also contribute significantly to our model’s final performance. For example, the IMU feature learning leads to a performance boost of 0.96% on Ego4D and 3.74% on EPIC-Kitchen, which is a good boost.
>
> On the other hand, as our method is, in essence, a distilled student model from the video teacher model with IMU information, it is natural that L_{KD} brings the largest performance boost.
>
> ---
>
>
> > 3. It could be better to compare the model performance in Table 4 (which is mismarked as Figure 4).
>
> Thanks for the suggestion! We have included a performance comparison in Table 4 as shown below:
>
> |   | GFLOPs | Runtime (ms) | Parameters (M) | Ego4D - Acc (%) | EPK - Acc (%) |
> | :--: | :--: | :--: | :--: | :--: | :--: |
> | Video | 369.51 | 10.7 | 108.91 | 46.38 | 77.28 |
> | AdaFuse | 15.2 | 2.04 | 38.85 | 36.35 | 48.15 |
> | STTS | 7.19 | 1.63 | 36.63 | 34.54 | 32.24 |
> | ListenToLook | 3.1 | 0.43 | 25.53 | 34.4 | 39.25 |
> | EgoDistill | 1.91 | 0.25 | 20.56 | 37.95 | 44.95 |
>
> We can observe from the table that EgoDistill requires significantly less computation while achieving good performance. Note that all the accuracy numbers here can be found in Figure 3 and Table 2 of our main paper; this table is simply to put all the information together in one place per the reviewer’s suggestion

---

### Official Review · Reviewer_K65f · 2023-07-04

**Soundness:** 4 excellent
**Presentation:** 4 excellent
**Contribution:** 4 excellent
**Rating:** 8
**Confidence:** 4

**Summary:**

The work presents a new approach and architecture, EgoDistill, which could efficiently perform action recognition through sparse video frames augmented with head motion data. The model goes through a two-stage training. A pre-training phase, which uses a contrastive loss that enables the motion encoder to learn semantic visual features, and a distillation phase, which makes the motion-visual fusion model jointly mimics the more stronger visual-only model. The results are impressive, supported by a series of ablation study.

There is no major flaws in the paper, except a minor confusion on the argument on the fact that the pre-training of the model forces the motion encoder learning semantic features.

**Strengths:**

The paper is very well written and easy to follow. The figure showcases the method very well.

The authors made a keen observation in using a separate modality (head motion) jointly with egocentric frames to efficiently perform action recognition task. The proposed architecture is novel and yields impressive results.

**Weaknesses:**

No major weaknesses.

**Questions:**

During the pre-training phase, L1 would train the fusion network as well as the IMU encoder, while NCE loss would mix and match samples to train both the IMU predictor and IMU encoder. How is the schedule of training the IMU predictor \hat{z} match with the optimization of the IMU encoder? How to prevent IMU encoder from being affected by under-trained IMU predictor?

**Limitations:**

I am not super convinced on the argument that the IMU encoder learns scene motion cues (line 318). The action confusion (between close/open and put/take) is a good example, but the two action pairs seem to be more relevant to time sequence rather than scene motion.

---

> ### Author Rebuttal · Authors · 2023-08-10
>
> Thank you for your careful review and constructive comments! We will address your comments point-by-point:
>
> ---
>
>
> > 1. During the pre-training phase, L1 would train the fusion network as well as the IMU encoder, while NCE loss would mix and match samples to train both the IMU predictor and IMU encoder. How is the schedule of training the IMU predictor \hat{z} match with the optimization of the IMU encoder? How to prevent IMU encoder from being affected by under-trained IMU predictor?
>
> That’s a good point!  We just initialize the IMU encoder and IMU predictor from scratch and train them directly with NCE loss and L1 loss.
>
> When both L1 loss and NCE loss is used, this will not be an issue as the IMU encoder will be trained by L1 loss, which encourages it to reconstruct the video model feature.
>
> When only NCE loss is used, we found that if we also finetune the image encoders, we would encounter the mode collapse problem, leading to a less useful IMU encoder. However, if we fix the image encoder and then train the IMU encoder and IMU predictor together, we observe that we are still able to achieve a useful IMU encoder (as shown in the 6th row of Table.1).
>
> We conjecture this is due the fact that the correlation between the starting and end image features (pretrained) and the IMU features are high enough such that the IMU predictor can be easily trained with NCE loss without mode collapsing.
>
> Nevertheless, your suggestion of using a special scheduling algorithm is very valuable. We will explore such mechanisms to further stabilize our self-supervised training stage.
>
> ---
>
>
> > 2. I am not super convinced by the argument that the IMU encoder learns scene motion cues (line 318). The action confusion (between close/open and put/take) is a good example, but the two action pairs seem to be more relevant to time sequence rather than scene motion.
>
>
> Actually, in this example, we think the time sequence and scene motion is the same: moving the door forward vs. pulling it back gives both different sequences of states as well as two different scene motions.
>
> Also, we note that it might be hard to intuitively understand from static figures in our main paper. Please refer to our **supplementary video** (04:50-05:55) for animated results. We hope this experiment could help explain what information does EgoDistill’s IMU encoder learn.

---

> > ### Comment · Reviewer_K65f · 2023-08-17
> > **Maintaining my evaluation**
> >
> > Thank the author for the response!
> >
> > After reading the review from other reviewers, I remain my position in supporting the author. I think the overall idea is straightforward and would make a nice addition to the community.

---

### Official Review · Reviewer_gfRs · 2023-07-06

**Soundness:** 3 good
**Presentation:** 3 good
**Contribution:** 3 good
**Rating:** 5
**Confidence:** 5

**Summary:**

This article presents an innovative approach called EgoDistill for egocentric video understanding. The authors highlight the computational challenges faced by existing models in this field and propose EgoDistill as a solution to overcome these limitations. EgoDistill aims to distill crucial information from egocentric video frames and IMU readings, resulting in a more efficient representation. The article provides detailed technical explanations of EgoDistill, illustrating how it effectively combines semantics from video frames with IMU readings. Additionally, the authors introduce a novel self-supervised training strategy specifically designed for IMU feature learning. To validate the effectiveness of EgoDistill, the authors conduct experiments and present compelling results. Their findings demonstrate that EgoDistill surpasses state-of-the-art efficient video understanding techniques when evaluated on the Ego4D and EPIC-Kitchens datasets. Overall, this article contributes significant advancements to the field of egocentric video understanding by introducing EgoDistill and showcasing its superior performance over existing methods.

**Strengths:**

1. EgoDistill offers several advantages over equivalent video models, including significantly reduced computational requirements in terms of GFLOPs, parameters, and inference time. This enhanced efficiency makes it a more practical option for real-world applications. Furthermore, the authors provide experimental evidence illustrating that EgoDistill surpasses state-of-the-art efficient video understanding methods when evaluated on the Ego4D and EPIC-Kitchens datasets.
2. The authors have devised a concise and effective self-supervised IMU feature learning architecture. This architecture enables the extraction of valuable camera motion features that are specifically associated with visual appearance changes. As a result, EgoDistill demonstrates improved performance in capturing and understanding these subtle visual cues, leading to enhanced overall performance.

**Weaknesses:**

1. The main text contains some basic errors, such as confusion between the order of figures and tables, as well as incorrect citations. It is crucial to carefully review and correct these issues.
2. Undoubtedly, EgoDistill exhibits commendable performance in terms of generalization and versatility. However, the explanation provided in the paper for why EgoDistill performs better than STTS when both methods incorporate IMU features (as mentioned in the "Effect of fusing IMU data" subsection) is too superficial and lacks a profound theoretical explanation for EgoDistill. Additionally, there is limited discussion on how the balance between different components of the model is controlled. It would be beneficial to address these points and provide more insights into the underlying mechanisms and strategies employed by EgoDistill.
3. EgoDistill relies on head motion data to capture camera pose changes. It is important to explore and discuss the limitations and potential mitigation strategies for scenarios where reliable head motion data may not be accurate.
4. The authors have not thoroughly discussed any potential shortcomings or limitations of EgoDistill. It would be valuable to identify specific scenarios or use cases where EgoDistill might not be as effective or efficient. Understanding these limitations would provide a more comprehensive understanding of the applicability and scope of EgoDistill and could open avenues for future research to address these areas of improvement.

**Questions:**

See Weaknesses

---

> ### Author Rebuttal · Authors · 2023-08-10
>
> Thank you for your careful review and constructive comments! We will address your comments point-by-point:
>
> ---
>
>
> > 1. The main text contains some basic errors, such as confusion between the order of figures and tables, as well as incorrect citations. It is crucial to carefully review and correct these issues.
>
> Thank you; we will correct them in our final version.
>
> ---
>
>
> > 2. Undoubtedly, EgoDistill exhibits commendable performance in terms of generalization and versatility. However, the explanation provided in the paper for why EgoDistill performs better than STTS when both methods incorporate IMU features (as mentioned in the "Effect of fusing IMU data" subsection) is too superficial and lacks a profound theoretical explanation for EgoDistill. Additionally, there is limited discussion on how the balance between different components of the model is controlled. It would be beneficial to address these points and provide more insights into the underlying mechanisms and strategies employed by EgoDistill.
>
> Please see L185-190 in the paper for discussion.  The reason that EgoDistill is able to outperform STTS+IMU is mainly that EgoDistill learns to jointly distill information from a video feature with *image and IMU* inputs. This distillation process will encourage the model to extract the interplay between static semantic information in the image and the dynamic information in IMU, as it is trained to reconstruct the video model feature. On the other hand, simple concatenation of the IMU feature to the image feature for action classification will not exploit the relationship between the image and IMU—it lacks the training signal to target the *dynamic video clip* feature—leading to inferior performance.
>
> To balance the different components of the model, we tune the weights in the loss function on the validation dataset for higher classification accuracy (L239-240).
>
> We will add the above notes to our final version.
>
>
> ---
>
>
> > 3. EgoDistill relies on head motion data to capture camera pose changes. It is important to explore and discuss the limitations and potential mitigation strategies for scenarios where reliable head motion data may not be accurate.
>
> As mentioned in L51-53, we use an inexpensive and widely available sensor for head tracking, which could already provide strong performance as we show.  Specifically,  the IMU sensor used to collect the datasets (Ego4D and EPIC-Kitchens) we use is pretty cheap ($3 per unit) and outputs reasonable but not perfect motion readings.
>
> Specifically, we observe that the raw IMU input has significant drifting and bias issues. This induces inconsistent correspondence between camera motion and IMU reading across different clips and videos. Therefore, for the IMU reading of each clip, on each dimension, we separately subtract raw readings by the mean values on the corresponding dimension. This operation normalizes IMU readings in each dimension to have zero average values. In this way, our model can only focus on the temporal motion patterns in each clip.  Learning-based methods used in inertial navigation and odometry works [1][2] may benefit any IMU model faced with low-quality IMU inputs. However, we find our simple mechanism already works well in our case.
>
> We will add this discussion to our final version of the paper.
>
> [1] RIO: Rotation-equivariance supervised learning of robust inertial odometry. Zhou et al.
> [2] RoNIN: Robust Neural Inertial Navigation in the Wild: Benchmark, Evaluations, and New Methods. Yan et al.
>
> ---
>
>
> > 4. The authors have not thoroughly discussed any potential shortcomings or limitations of EgoDistill. It would be valuable to identify specific scenarios or use cases where EgoDistill might not be as effective or efficient. Understanding these limitations would provide a more comprehensive understanding of the applicability and scope of EgoDistill and could open avenues for future research to address these areas of improvement.
>
> We agree that discussion of limitations of EgoDistill is important.
>
> Although EgoDistill is useful for many action categories in egocentric videos, for some other classes it is less effective. To have a clearer understanding EgoDistill’s limitations, we analyze when IMU is useful and its limitations:
>
> (Please refer to the attached **PDF** file in the overall author rebuttal for the mentioned figures.)
>
> Here, we explore what kinds of clips are better reconstructed using EgoDistill. Figure.A1 in the uploaded pdf file shows the improvement of EgoDistill over the VisOnly-Distill model on Ego4D and EPIC-Kitchens split by action class.  We observe that IMU is more useful for actions with predictable head motion (e.g., *break*, *cut*, *close*), and is less helpful for actions where head motion may be small or unrelated (e.g., *empty*, *fill*, *press*).
>
> Figure.A2 in the uploaded pdf file shows clip examples whose video features are best and worst reconstructed.  We observe that the best-reconstructed clips (top) contain moderate head motion that is predictive of scene motion and action semantics. For example, the camera wearer's head moves slightly backward while opening the cabinet.  On the other hand, more poorly reconstructed clips tend to contain little head motion (third row)---in which case IMU is redundant to the RGB frame---or drastic head motion that is weakly correlated with the camera wearer's activity and introduces blur to the frame (last row).
>
> Additionally, EgoDistill might not be useful when we want to focus on the “noun” part of the action (e.g., differentiating watching TV and using the cell phone), as IMU information does not contain information about the object semantics in the scene.
>
> We will add this discussion to our final version of the paper about the scope and applicability of EgoDistill.

---

### Official Review · Reviewer_xst2 · 2023-07-06

**Soundness:** 3 good
**Presentation:** 3 good
**Contribution:** 3 good
**Rating:** 5
**Confidence:** 4

**Summary:**

This paper for on Ego-centric video understanding, a promising and popular topic recently.
Different from previous works that focus on improve heavy video encoder, this work try to cut the computation cost down and focus on lightweight temporal reasoning.

Please reply the questions and weakness point-by-point.

**Strengths:**

- The motivation is quite good and make sense. I like it. Actually ego-centric video have fixed background in general and learn the motion pattern based on the first frame is more easily than third-view video. Considering this area is  new and not well studied, I think it's valuable.
- The method is also simple but works.
- The presentation is good and I can understand it very well.
- The computation cost is cut a lot and provide possibly for small and mobile device.


**Weaknesses:**

- The method section especially the IMU predictor is not solid. Actually, the idea of bring GRU for temporal modeling is quite a old technique. There also exist a lot of works that predict the temporal sequence on feature level like MemDPC two years ago.
- Although the performance is not a very important measurement to me, the compared method is somewhat old. For example, Line243-248.

**Questions:**

- Why the first row of Fig.6 sometimes seems not well matched with the action, can you give detail explanation about how to get this figure and why always no changed? The head motion is from AR glasses or?
-  How to define ambiguous categories  in Ego4D?


**Limitations:**

- The performance is hard to compare with video encoder based method.
- The proposed work also suitable for ego-centric video.

---

> ### Author Rebuttal · Authors · 2023-08-10
>
> Thank you for your constructive comments and positive feedback! We will answer your questions point-by-point:
>
> ---
>
>
> > The method section especially the IMU predictor is not solid. Actually, the idea of bring GRU for temporal modeling is quite a old technique. There also exist a lot of works that predict the temporal sequence on feature level like MemDPC two years ago. Although the performance is not a very important measurement to me, the compared method is somewhat old.
>
> We regard the GRU as a simple and effective architecture for aggregating frame information in our setting, as demonstrated in the results (see Figure 3 where we observe significant performance gain with more frames).
>
> We discuss our choice of baselines in L249. To summarize, these methods represent recent, leading approaches in the area of efficient video recognition (e.g., STTS, ECCV22). Recent architectures are promising for performance but are far less efficient, making them unsuitable given our focus on efficiency.
>
> ---
>
>
> > 1. Why the first row of Fig.6 sometimes seems not well matched with the action, can you give detail explanation about how to get this figure and why always no changed? The head motion is from AR glasses or?
>
> We obtain the head motion from the IMU sensor of a GoPro camera, which is the device used to collect data in these egocentric video datasets. To obtain the figure, we double integrate the IMU readings (which are accelerations) to obtain the motion trajectory of the head. Then, by setting the initial head position at the origin and facing the x-axis, we plot the head trajectory across time.
>
> With only static figures it can be hard to match the head motion plot in Fig.6 with the action shown in the video frames. Therefore, we refer the reviewer to our **supplementary video** to watch an animated version of Fig.6 (from 03:00 to 04:11). We hope the animated version makes the correspondence clearer.
>
> ---
>
>
> > 2. How to define ambiguous categories in Ego4D?
>
> We define ambiguous categories in Ego4D as the ones that cannot be determined by only a few frames (L318-320). For example, only from a single frame of the video clip, one cannot distinguish between “close” and “open”. By using IMU signals to obtain head motions, our model is able to classify these two categories despite having only a few (1-4) visual frames.
>
>
> To have a clearer understanding of the ambiguous categories, we analyze when IMU is useful and its limitations:
>
> (Please refer to the attached **PDF** file for the mentioned figures.)
>
> We saw in our paper that using IMU leads to an overall performance improvement in action recognition, indicating better video feature prediction capability  (Lines 265-272 and Figure 3).
>
> Here, we explore what kinds of clips are better reconstructed using EgoDistill. Figure.A1 in the uploaded pdf file shows the improvement of EgoDistill over the VisOnly-Distill model on Ego4D and EPIC-Kitchens split by action class.  We observe that IMU is more useful for actions with predictable head motion (e.g., *break*, *cut*, *close*), and is less helpful for actions where head motion may be small or unrelated (e.g., *empty*, *fill*, *press*).
>
> Figure.A2 in the uploaded pdf file shows clip examples whose video features are best and worst reconstructed.  We observe that the best-reconstructed clips (top) contain moderate head motion that is predictive of scene motion and action semantics. For example, the camera wearer's head moves slightly backward while opening the cabinet.  On the other hand, more poorly reconstructed clips tend to contain little head motion (third row)---in which case IMU is redundant to the RGB frame---or drastic head motion that is weakly correlated with the camera wearer's activity and introduces blur to the frame (last row).
>
>
> ---
>
>
> > 3. The performance is hard to compare with video encoder based method.
>
> The purpose of EgoDistill is to obtain an efficient video understanding method for egocentric videos. In this regard, EgoDistill reaches a significantly better performance-efficiency balance than the compared methods. Compared with video-based methods (e.g., MotionFormer), our frame-based model requires significantly less computation resources, which is vital for mobile devices.

---

### Official Review · Reviewer_M5sy · 2023-07-07

**Soundness:** 3 good
**Presentation:** 3 good
**Contribution:** 3 good
**Rating:** 6
**Confidence:** 4

**Summary:**

The work studies efficient video recognition via distilling IMU signals. Instead of heavily processing each frame, the authors propose to leverage very sparse frames (1-4) for visual features and IMU data. It leads to much more efficient inference cost. The proposed method has shown improvement over other multi-modality fusion techniques like AdaFuse, etc. They also show IMU trajectory provides complementary information to sparse frames.

**Strengths:**

+ The idea to leverage IMU signals for efficiency is novel. And the empirical results support the idea. The improvement with IMU is consistent across different architectures.
+ The two stage training (first aligns IMU features then fusing them) is shown effective over alternative multi-modality fusion baselines.
+ The claims are well supported by ablation studies like two-stage training, loss term, usage of modality, etc.



**Weaknesses:**

1. Since IMU only represents head motions, it is sensible to disambiguate activity relevant to scene motion (open / close, put /take), but it may not contain enough signals about finer grained activity temporally or spatially. But it is the problem of IMU signal, not the paper’s.
2. I wonder how much does the motion _trajectory_ help, compared to the overall displacement between the end frame and the starting frame, i.e X_t - X_0. More specifically, in table 2, how does EgoDistill with less IMU steps work and how does feeding start frame __and__ end frame works?


**Questions:**

See weakness above

**Limitations:**

Limitation is discussed

---

> ### Author Rebuttal · Authors · 2023-08-10
>
> Thank you for your constructive comments! We will address your concerns point-by-point:
>
> ---
>
> > 1. Since IMU only represents head motions, it is sensible to disambiguate activity relevant to scene motion (open / close, put /take), but it may not contain enough signals about finer-grained activity temporally or spatially. But it is the problem of IMU signal, not the paper’s.
>
> Yes, although IMU is useful for many action categories in egocentric videos, it cannot provide much information about some other categories.
>
> To understand how information in IMU contributes to the performance boost, we analyze when IMU is useful and its limitations:
>
> (Please refer to the attached **PDF** file in the overall author rebuttal for the mentioned figures.)
>
> We saw in our paper that using IMU leads to an overall performance improvement in action recognition, indicating better video feature prediction capability (Lines 265-272 and Figure 3). Here, we explore what kinds of clips are better reconstructed using EgoDistill. Figure.A1 in the uploaded pdf file shows the improvement of EgoDistill over the VisOnly-Distill model on Ego4D and EPIC-Kitchens split by action class.  We observe that IMU is more useful for actions with predictable head motion (e.g., *break*, *cut*, *close*), and is less helpful for actions where head motion may be small or unrelated (e.g., *empty*, *fill*, *press*).
>
> Figure.A2 in the uploaded pdf file shows clip examples whose video features are best and worst reconstructed.We observe that the best-reconstructed clips (top) contain moderate head motion that is predictive of scene motion and action semantics. For example, the camera wearer's head moves slightly backward while opening the cabinet.On the other hand, more poorly reconstructed clips tend to contain little head motion (third row)---in which case IMU is redundant to the RGB frame---or drastic head motion that is weakly correlated with the camera wearer's activity and introduces blur to the frame (last row).
>
> ---
>
> > 2. I wonder how much does the motion trajectory help, compared to the overall displacement between the end frame and the starting frame. More specifically, in table 2, how does EgoDistill with less IMU steps work and how does feeding start frame and end frame works?
>
> This is a good point! To address your questions, we provide two sets of experiments.
>
> **Firstly, we conduct an ablation study of feeding different input frames**:
>
> * N=1 - use the center frame
> * N=2 - **use the start frame and end frame**
> * N=4 - uniformly sample 4 frames
>
>
> We compare the performance of EgoDistill and VisOnly-Distill under different settings:
>
> | Input Frame | Ego4D |   | EPIC-Kitchens |   |   |
> | :--: | :--: | :--: | :--: | :--: | :--: |
> |   | EgoDistill | VisOnly-D | EgoDistill | VisOnly-D | GFLOPs |
> | N=1 (center) | 37.95 | 34.32 | 44.95 | 37.2 | 1.9 |
> | N=2(first and last) | 38.29 | 34.48 | 50.07 | 46.71 | 3.8 |
> | N=4 (uniform) | 38.46 | 34.84 | 52.43 | 48.96 | 7.6 |
>
> From the results above, we observe that only using the overall displacement between the end frame and the starting frame (N=2) leads to indeed better performance of VisOnly-D than using only one frame (N=1). However, the motion trajectory still helps EgoDistill obtain significantly better performance in either case. This result indicates the information contained in the motion trajectory captured by IMU.  N=2, 4 leads to better performance, but requires higher computational cost as noted in L236.
>
> **Secondly, we conduct an ablation study about using fewer IMU steps**.
>
> According to the suggestion, we experiment with feeding EgoDistill with partial IMU readings of the full video clip. Specifically, we train and evaluate our model with only the first N% IMU reading data of the full clip and pad the rest with 0. Results are below
>
> | IMU Usage | Ego4D | EPIC-Kitchens |
> | :--: | :--: | :--: |
> | 100% IMU | 37.95 | 44.95 |
> | 75% IMU | 37.88 | 43.43 |
> | 50% IMU | 37.65 | 43.69 |
> | 25% IMU | 37.21 | 42.68 |
> | 10% IMU | 36.6 | 41.45 |
> | No IMU (VisOnly-D) | 34.32 | 37.2 |
>
> This ablation study shows that more complete IMU readings yield the best results, while using 25% and 50% IMU data achieves the majority of the gain over the “No IMU” baseline.

---

### Author Rebuttal · Authors · 2023-08-10

We sincerely thank all the reviewers for their hard work and constructive feedback!

Four reviewers recommend accepting (one as a Strong Accept).

Overall the reviewers give very positive comments on our work, e.g., “contributes significant advancements to the field of egocentric video understanding”, “The motivation is quite good and make sense. I like it.”, “The authors made a keen observation in using a separate modality.”,  “The proposed architecture is novel and yields impressive results.” “Claims are well supported”, “novel and yields impressive results”, “much more efficient”, “presentation is good and I can understand it very well”, “very well written and easy to follow”, “innovative approach”, “surpasses state-of-the-art efficient video understanding”, “no major weaknesses”, We thank them for their appreciation.

We address questions from different reviewers separately. We upload a **pdf** file in the overall rebuttal for figures used to reply to some of the reviewers’ questions.

---

### Decision · Program_Chairs · 2023-09-21

**Decision:**

Accept (poster)

**Comment:**

Four of the five reviewers recommend acceptance; the authors reasonably address the fifth reviewer's concerns, although the reviewer did not respond.  Based on the reviews and responses, the AC recommends accepting the paper.  Authors are recommended to revise the work based on the reviewers' suggestions.